# The differential burden of acute rhinovirus infections in children with underlying conditions

María Isabel Sánchez Códez[1], Isabel Benavente Fernández[2,3,4], Katherine Moyer[5], Amy L. Leber[6], Octavio Ramilo[7], Asuncion Mejias [7]*

1 Division of Pediatric Infectious Diseases, Puerta del Mar, Cadiz, Spain, 2 Department of Paediatrics, Puerta del Mar University Hospital, Cadiz, Spain, 3 Biomedical Research and Innovation Institute of Cádiz (INiBICA) Research Unit, Spain, 4 Area of Paediatrics, Department of Child and Mother Health and Radiology, Medical School, University of Cadiz, Spain, 5 Division of Pediatric Infectious Diseases, Inova Children's Hospital, Falls Church, Virginia, United States of America, 6 Department of Laboratory Medicine, Nationwide Children's Hospital, Columbus, Ohio, United States of America, 7 Department of Infectious Diseases, St Jude Children's Research Hospital, Memphis, Tennessee, United States of America

* asuncion.mejias@stjude.org

## Abstract

### Introduction

Rhinoviruses (RVs) are a well-known trigger of wheezing episodes in children with asthma. Their role in other pediatric chronic medical conditions is not fully known.

### Methods

Patients ≤21 years hospitalized or evaluated as outpatients with symptomatic RV infection were identified from 2011–2013. Patients were categorized based on the type of underlying disease and differences in clinical parameters, RV loads ($C_T$ values), viral, bacterial and fungal co-infections and clinical outcomes were compared between groups. Multivariable analyses were performed to identify the comorbidities associated with oxygen requirement, PICU admission, and prolonged hospitalization.

### Results

Of 1,899 children identified, 77.7% (n = 1477) had an underlying comorbidity including asthma/atopy (36.8%), prematurity (7.7%), chronic respiratory diseases (6.4%), congenital heart disease (CHD, 3.2%), immunocompromised hosts (ICH; 1.4%) and others (22.2%). Prevalence of comorbidities increased with age (70%, infants *vs* 84%-87%, children >1 year; p < 0.0001). Median RV loads were intermediate-high (24–26 $C_T$ values), irrespective of the underlying disease. RV/viral co-detections were identified in 11% of ICH vs 20%- 30% in all other children whereas bacterial co-infections were identified in 2.9% of children. Multivariable models identified asthma/atopy, prematurity, CHD and bacterial co-infections consistently associated

**Data availability statement:** The minimal dataset supporting the findings of this study is publicly available and can be accessed at Harvard Dataverse (DOI: https://doi.org/10.7910/DVN/VPPNQA).

**Funding:** The author(s) received no specific funding for this work.

**Competing interests:** I have read the journal's policy and the authors of this manuscript have the following competing interests: AM has received fees for participation in Advisory Boards from Janssen, Merck, Pfizer, Moderna, Enanta, AstraZeneca and Sanofi-Pasteur and grants from NIH, Merck and Janssen to institution. OR has received fees for participation in Advisory Boards from Merck, Pfizer, AstraZeneca and Sanofi-Pasteur and grants from NIH, Bill & Melinda Gates Foundation, Merck and Janssen to institution. ALL has received research grants from BioFire, Cepheid, Luminex, and Diasorin, and consulting fees from Medscape, Biorad and BioFire. The remaining authors have no conflicts of interest.

with all three clinical outcomes (p < 0.0001). Older age and higher RV loads were also associated with increased odds of PICU admission.

## Conclusions

The prevalence of comorbidities was high in children with RV infections. Of those, asthma/atopy, prematurity and CHD were consistently associated with severe disease. Higher RV loads and bacterial co-infections, although infrequent, were also associated with worse clinical outcomes, suggesting the importance of defining clinical phenotypes for future targeted interventions.

## Introduction

Rhinoviruses (RVs) are a leading cause of upper and lower respiratory tract infections (LRTI) in children and adults [1,2]. The causative role of RVs in respiratory diseases has been underestimated as they are commonly detected in asymptomatic subjects and associated with prolonged shedding [3–6]. Nonetheless, RVs have been recognized as important triggers of wheezing episodes in asthmatic children [7]. In addition to asthma, pediatric studies suggest that certain underlying conditions, such as immunocompromised hosts (ICH), prematurity, or congenital heart disease (CHD), have an increased risk for severe RV infections (ARI) [1,8–11]. Whether the increased risk for severe RV disease extends to children with other comorbidities, and whether specific clinical parameters or co-detection with other pathogens are associated with increased morbidity is not fully known.

The aim of the present study was to define the clinical phenotype and the differences in clinical presentation, healthcare utilization and the impact of RV loads and of co-infections in a large cohort of pediatric patients with RVs infection according to the type of underlying condition.

## Materials and methods

### Patient population and study design

This was a retrospective study including a convenience sample of children and adolescents ≤21 years of age with symptomatic RVs infection diagnosed in the inpatient setting or as outpatients from 7/2011–12/2013. We included patients up to 21 years of age since this was the catchment population that is cared for at Nationwide Children's Hospital (NCH), a common practice in US pediatric hospitals [12]. Children and adolescents were identified by virology reports, clinical was data extracted from electronic healthcare records (EHR), and manually reviewed by two independent investigators. Children with incomplete data and non-respiratory diagnoses were excluded. To ensure that the episodes analyzed were distinct, duplicate encounters during the same calendar year were also excluded, as RV can be shed for months after the acute episode. The present study involves the analysis of a subset of children as part of a larger cohort that has been previously published [13].

RV testing was performed per standard of care in respiratory samples using a RV rt-PCR targeting the 5′NCR (or 5′UTR) region as described [14,15]. RV molecular typing was not performed. Semiquantitative RV viral loads were reported as cycle threshold ($C_T$) values, with a $C_T$ value > 40 being considered not detected. RV testing was run in parallel with individual PCRs for respiratory syncytial virus (RSV), parainfluenza virus (PIV), influenza virus A/B, adenovirus (ADV), and human metapneumovirus (hMPV) as part of a panel [16]. EHR were assessed for pertinent demographic and clinical data including the admission diagnosis, laboratory and radiologic findings and co-infections. Viral co-detections were defined as the simultaneous detection by PCR of one of the other respiratory viruses included in the panel. Bacterial or fungal co-infections were defined as the isolation of these pathogens by culture from a sterile site (blood, pleural fluid) or the lower respiratory tract. In addition, in symptomatic children with cystic fibrosis (CF), throat culture samples were considered valid for detecting both bacterial and fungal pathogens. We collected information regarding healthcare utilization including medications, need for hospitalization, supplemental oxygen, PICU care and total length of stay. For analysis purposes children were grouped into seven categories based on the type of underlying disease: asthma/atopy; other chronic respiratory conditions including chronic lung disease (CLD) or CF; prematurity without CLD; CHD; ICH; others (i.e., endocrine, renal, etc); and none (previously healthy). Children with more than one comorbidity were included in the disease category that had the greatest impact on RV disease severity based on previous studies [8–11,17], such as that a child with asthma/atopy and renal disease was categorized under asthma/atopy. The hierarchical order used for children with more than one underlying condition was as follows: ICH; asthma/atopy; other chronic respiratory conditions; prematurity without CLD; CHD and others (endocrine, renal, etc).

The study was approved by NCH Institutional Review Board (IRB#13–00796). It was granted exemption from obtaining informed consent due to its retrospective design in accordance with the US Department of Health and Human Services regulations, specifically 45 CFR 46. Data was accessed from EHR over a 11-month period (February 6th through December 19th of 2014), and any identifier linked to participant individuals destroyed upon data collection and curation.

## Statistical analysis

Qualitative data were expressed as percentages, and quantitative variables as medians and 25%-75% interquartile range (IQR) as data were not normally distributed. Chi-squared test or Fisher's exact tests were used to compare proportions, and Kruskal–Wallis tests for continuous data followed by the Holm post-hoc test to adjust for multiple comparisons. In addition, the Bonferroni correction was applied to correct for multiple testing when analyzing families that included multiple parameters such as diagnoses, medications used or underlying conditions. Multivariable logistic regression was performed to define the risk for supplemental oxygen administration, PICU admission and longer length of stay according to the type of comorbidity, adjusted by age, sex, RV loads and viral and bacterial co-infections. For analysis purposes length of stay was dichotomized by the median which was 2.1 days. Two-sided p-values <0.05 were considered statistically significant. Analyses were conducted using STATA 16.0. (StataCorp, College Station, TX, USA).

## Results

### Demographic and clinical characteristics

Of the 1,899 children with symptomatic RV infection, 77.7% (n = 1,477) had underlying comorbidities. Asthma/atopy was the most prevalent (36.8%; 699) followed by prematurity (7.7%; 147); other chronic respiratory diseases (6.4%; 121); CHD (3.2%; 61); and ICH (1.4%; 27). Chronic respiratory diseases included CF (0.9%; 18), CLD (3.2%; 61), and less common conditions such as tracheoesophageal fistula, pulmonary hypertension, spontaneous pneumothorax, airway malacia, subglottic stenosis, cleft palate, and Pierre-Robin sequence. The remaining children with other comorbidities (22.2%; 422) included genetic syndromes (6.9%; 132), gastrointestinal diseases (3.6%; 68), neurologic disorders (1.9%; 36),

hematologic diseases (0.9%; 17), and a variety of other conditions, including endocrine, osteoarticular and genitourinary abnormalities (8.9%; 169) (S1 Table **in** S1 File).

Median age for children with comorbidities (15.8 [4.9–52.9] months) was greater than for previously healthy (4.7 [1.5–19.9] months, p = 0.0001); **Table 1**.

ICH were the oldest group (68.6 [42.5–116.3] months). The prevalence of comorbidities increased after the first year of life, when 84% to 87% of children had a chronic medical condition vs 70% in infants (p < 0.0001; **Fig 1**).

The prevalence of asthma/atopy doubled in children >1 year (45%-57%) vs infants (26%; p = 0.007). Prematurity was more common in infants and children 1–5 years (10% and 7.6% respectively) than in older children (1.5%-2.5%; p = 0.007). Only 0.2% of infants were ICH vs 4.6% of children 11 to < 21 yr (p = 0.007; S2 Table **in** S1 File). Upper respiratory infection (URI) was the most common diagnosis in ICH (55.6%) and healthy children (48.3%) while wheezing

**Table 1. Demographics, laboratory, radiology parameters and distribution of diagnoses of children and adolescents with RV infection.**

| | Previously healthy (n = 422; 22.2%) | Asthma/Atopy (n = 699; 37%) | Chronic resp conditions † (n = 121; 6.4%) | Prematurity (n = 147; 7.7%) | CHD (n = 61; 3.2%) | ICH (n = 27; 1.4%) | Others § (n = 422; 22.2%) | p value |
|---|---|---|---|---|---|---|---|---|
| **Demographic characteristics** | | | | | | | | |
| **Age,** (months) | 4.7 [1.5-19.9] ●○ | 23 [8.2-68.5] ●○ | 11.6 [5.5-21.2] ●○■□ | 6.7 [2.6-18.7] ○■□ | 14.7 [3.5-53.4]●■□ | 68.6 [42.5-116.3]●○■ | 10.5 [3.1-36.8]●○■□ | 0.0001 |
| **Sex,** n (%) male | 243 (57.6) | 416 (59.5) | 65 (53.7) | 87 (59.2) | 44 (72.1) | 19 (70.4) | 263 (62.3) | 0.174 |
| **Laboratory/Radiology data** | | | | | | | | |
| **WBC** (cells/ul) | 10,900 [8,400-14,900]●○ | 13,500 [9,600-17,200]●○□ | 11,700 [8,600-15,700]● | 11,100 [7,900-16,000]● | 10,600 [8,900-16,700]● | 2,000 [1,400-3,600]● | 11,200 [7,900-16,500]●□ | 0.0001* |
| **ALC** (cells/ul) | 4,251 [2,180-6,431]○ | 2,025 [1,080-4,590]○ | 2,925 [2,223-5,499]○ | 4,051 [2,220-5,772]○ | 3,816 [848-6,254]○ | 840 [500-1,260]○ | 5,488 [3,248-7,280]○ | 0.0001* |
| **ANC** (cells/ul) | 5,100 [2,700-9,700]○ | 10,700 [6,900-15,200]○ | 7,600 [4,300-11,300] | 6,100 [11,900-12,500]○ | 6,900 [4,200-13,800] | 2,100 [300-3,900]○ | 6,100 [2,900-11,500]○ | 0.0001* |
| **Chest X-ray performed,** n (%) | 236 (55.9)○ | 541 (77.4)○□ | 94 (77.7)○ | 111 (75.5)○ | 49 (80.3)○ | 16 (59.3) | 289 (68.5)○□ | 0.0001* |
| **Chest X-ray abnormalities,** n (%) | 181/236 (76.7) | 442/541 (81.7)○ | 80/94 (85.1)□ | 83/111 (74.8) | 41/49 (83.7) | 8/16 (50.0)○□ | 233/289 (80.6) | 0.017* |
| **Diagnosis** | | | | | | | | |
| **URI** | 204 (48.3)● | 119 (17.0)●■ | 14 (11.6)●▲ | 35 (23.8)●○♦ | 19 (31.1) | 15 (55.6)■▲○ | 188 (44.5)■▲♦ | 0.0001* |
| **Fever** | 8 (1.9) | 9 (1.3) | 0 (-) | 2 (1.4) | 2 (3.3) | 1(3.7) | 7 (1.7) | 0.643 |
| **AOM/sinusitis** | 9 (2.1) | 3 (0.4) | 0 (-) | 1 (0.7) | 0 (-) | 1 (3.7) | 8 (1.9) | 0.998 |
| **Wheezing illnesses** | 102 (24.2)●○ | 443 (63.4)●○ | 51 (42.2)●○□ | 73 (49.7)○ | 22 (36.1)○ | 7 (25.9)●○ | 116 (27.5)○□ | 0.0001* |
| **PNA/LRTI** | 78 (18.5)○ | 94 (13.5)○ | 51 (42.2)○ | 21 (14.3)○ | 14 (23.0)○ | 2 (7.4)○ | 72 (17.1)○ | 0.0001* |
| **Respiratory Failure** | 21 (4.9) | 31 (4.4) | 5 (4.1) | 15 (10.2) | 4 (6.6) | 1(3.7) | 31 (7.3) | 0.087 |

Continuous variables are expressed as medians 25–75% interquartile ranges (IQR) and categorical data as frequency (%). Continuous variables were analyzed by Kruskal-Wallis test followed by Holm post-hoc test. Categorical data were analyzed by χ² test. ALC: absolute lymphocyte counts; ANC, absolute neutrophil count; AOM, acute otitis media; CHD, congenital heart diseases; ICH, immunocompromised host; LRTI, lower respiratory tract infection; PNA, pneumonia; URI, upper respiratory infection; WBC: white blood cell. Asterisks (*) indicate significant p values after applying the Bonferroni correction to adjust for multiple testing for families that included multiple parameters. Significant *p values* <0.001 are represented by dark symbols (●, ■, ▲, ♦) and significant *p values* <0.05 are represented by blank symbols (○, □, △, ◇). Matching symbols indicate the groups for which significant differences were identified.

†Chronic respiratory conditions included: cystic fibrosis (CF) (n = 18); chronic lung disease (CLD) (n = 61); tracheoesophageal fistula, pulmonary hypertension, spontaneous pneumothorax, airway malacia, subglottic stenosis, cleft palate, and Pierre–Robin sequence.

§Others included: genetic syndromes (n = 132); gastrointestinal diseases (n = 68); neurological disorders (n = 36); hematologic disease (n = 17) and varied including endocrine, osteoarticular, and genitourinary underlying diseases (n = 169).

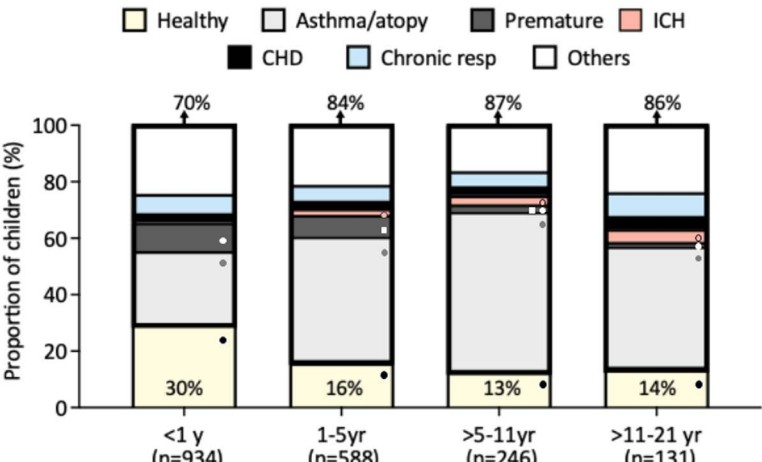

**Fig 1. Distribution of underlying conditions according to age.** The Y axis depicts the proportion of children and adolescents with RV ARI and the X axis the different age groups, with the number of patients included in each age group underneath the bars. Percentages at the top of the bars indicate the overall proportion of children with underlying conditions for that age group, and those at the bottom the proportion of previously healthy children per age group, that are color-coded in yellow. The different underlying comorbidities are represented as follows: grey (asthma/atopy), dark grey (premature), red (ICH), black (CHD), blue (chronic resp diseases) and white (others). CHD, congenital heart diseases; Chronic resp, other chronic respiratory conditions; ICH, immunocompromised host; yr, years. Analyses by χ2 test. Significant *p values* <0.05 values are represented by symbols (●, ■). Matching symbols indicate the groups for which significant differences were identifed.

illnesses predominated in children with asthma/atopy (63.4%) and prematurity (49.7%). Pneumonia was the initial diagnosis in 42.2% of children with chronic respiratory conditions and in 18.5% of healthy children (p = 0001). Fever, acute otitis media or sinusitis represented <4% of the admitting diagnoses.

In terms of laboratory data, 52.4% (995/1899) of children underwent blood work-up. Leukocytosis was more common in asthmatic/atopic children while leukopenia predominated in ICH. Abnormal radiologic findings were reported in 50% of ICH children *vs* 75%-85% of children with other comorbidities (p = 0.017). The most common radiologic patterns were bronchial wall thickening and atelectasis in 56.0% to 67.2% of children respectively (S3 Table **in** S1 File).

### RV loads and viral and bacterial co-infections

Semiquantitative RV loads ($C_T$ values) were similar between healthy (25.6 $C_T$ values) and children with comorbidities (25.05 $C_T$ values), irrespective of the type of comorbidity (S4 Table **in** S1 File). A concomitant respiratory virus was identified in 26.5% (112/422) of healthy children and 19.8%-29.9% (347/1477) of those with comorbidities (**Fig 2A**, S4 Table **in** S1 File). ADV and RSV followed by PIV were the most common co-detected viruses. RV/ADV co-detection was more common in asthmatic/atopic (8.9%) and premature children (11.6%; p = 0.05), whereas RV/RSV co-detections were more frequent in CHD (18.0%), chronic respiratory conditions (7.4%) and children with other comorbidities (7.8%). In healthy children, rates of RV/ADV or RV/RSV co-detection were comparable and ~9%. Co-detection of other respiratory viruses in ICH was rare and identified in only three children (RV/RSV, n = 2; RV/PIV, n = 1). More than one additional respiratory virus was identified in 2.8% (n = 54) of children and the most frequent combination was RV/ADV/RSV in two-third of cases.

Bacterial co-infections were identified in 56 (2.9%) children based on lower respiratory/BAL cultures (n = 48), throat cultures from patients with CF (n = 4), or blood cultures (n = 4) and were more common in children with chronic respiratory conditions (9.9%) and ICH (7.4%); p = 0.001; (S4 Table **in** S1 File, **Fig 2B**). The most common bacteria identified were *S. pneumoniae, S. aureus and P. aeruginosa.* The proportion of gram-positive bacteria was greater in asthmatic/atopy children (58.3%) while gram-negative bacteria were mor commonly cultured in children with chronic respiratory conditions (71.4%). Differences did not reach statistical significance.

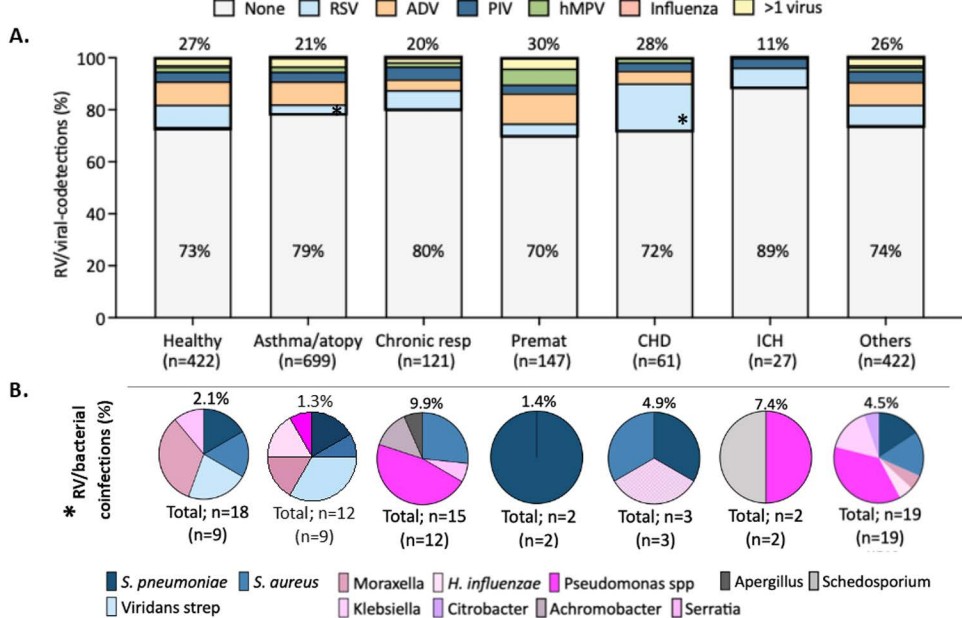

**Fig 2. RV/viral co-detections and RV/bacterial/fungal co-infections in children and adolescents with RV infections. (A)** Bars represent the proportion of viral co-detections in children with RV infection in the different disease groups (healthy, asthma/atopy, chronic respiratory diseases, prematurity, CHD, ICH and others). The proportion of RV-single infections are depicted inside the bars colored in grey, and the percentages of RV/viral co-detections are above each bar. RV/RSV is color coded in blue, RV/ADV in orange, RV/PIV in dark blue, RV/hPMV in green and RV/>1 respiratory virus in yellow. Analyses by χ2 test. The asterisk indicates the two groups that were significantly different showing that children with CHD had significantly higher rates of RSV infection than those with asthma/atopy ($p<0.01$). **(B)** Pie charts represent the proportion of the different microorganisms isolated in each group with the total percentage of bacterial/fungal co-infections identified include above each pie chart. The total number of pathogens identified is depicted underneath the pies along with the number of patients (in parenthesis). Pink and blue colors indicate gram-negative and gram-positive bacteria respectively. Grey color indicates fungal pathogen. Analyses by χ2 test. For bacterial co-infections significant differences were identified between children with chronic respiratory conditions vs. all the type of comorbidities except CHD and others ($p<0.05$). RV, rhinovirus, RSV, respiratory syncytial virus, ADV, adenovirus; hMPV, human metapneumovirus; PIV, parainfluenza virus; CHD, congenital heart diseases; ICH, immunocompromised host; Premat, premature; resp, respiratory.

## Healthcare utilization *in* children *with* RV ARI

We assessed the use of medications and other indicators of healthcare utilization (**Table 2**).

Systemic steroids were used more commonly in children with asthma/atopy, chronic respiratory conditions and prematurity (41.5–62.4%) than in healthy children (20.7%; $p=0.0001$). Bronchodilators including albuterol, ipratropium bromide or magnesium sulfate, were administered to~75% of children with asthma/atopy and other chronic respiratory diseases and in 14.8% of ICH, with no differences between groups (p>0.05). Antibiotics were administered to~40% of children, and their use was significantly higher in ICH (67%; $p<0.001$).

Overall, 86% of children with comorbidities were hospitalized vs 77% of previously healthy ($p=0.0001$). The number of children who required supplemental oxygen was also higher in children with comorbidities (p<0.001). This difference was mainly driven by children with asthma/atopy (65.2%), CHD (65.6%) and other chronic respiratory diseases (65.3%) vs healthy (35%) or ICH (18.5%). PICU admission was higher in children with comorbidities (34%) vs healthy (17%) or ICH (11%); $p=0.0001$. This difference was also driven by children with asthma/atopy and CHD. Children with chronic respiratory diseases, CHD, prematurity or other underlying diseases required mechanical ventilation more commonly (11%-16%) than children with asthma/atopy, ICH or those previously healthy (<5%). Total length of stay was shorter for healthy children than those with comorbidities (1.7 vs 2.4 days; p=0.0001).

**Table 2. Healthcare utilization in children and adolescents with RV infection.**

| | Previously healthy (n=422) | Asthma/Atopy (n=699) | Chronic respiratory conditions † (n=121) | Prematurity (n=147) | CHD (n=61) | ICH (n=27) | Others § (n=422) | p value |
|---|---|---|---|---|---|---|---|---|
| **Medications** | | | | | | | | |
| **Systemic steroids,** n (%) | 87 (20.7)●○ | 436 (62.4)●○ | 58 (47.9)●○■ | 61 (41.5)●○■ | 21 (34.4)○ | 6 (22.2)○ | 90 (21.3)○■ | 0.0001* |
| **Steroids duration** (days) | 2 [1–3]● | 2 [1–3] | 3 [2–4]● | 2 [1–3] | 3 [2–3] | 3.5 [2–8] | 2 [1–4] | 0.002* |
| **Bronchodilators,** n (%) | 167 (39.6) | 535 (76.5) | 91 (75.2) | 94 (63.9) | 37 (60.7) | 4 (14.8) | 200 (47.4) | 0.650 |
| **Bronchodilator duration** (days) | 2 [1–4] | 1 [1–2]● | 2 [1–4] | 2 [1–2] | 2 [1–7] | 2 [1–3] | 2 [1–4]● | 0.005* |
| **Antibiotics,** n (%) | 185 (43.8) | 246 (35.2)○ | 55 (45.5) | 63 (42.9) | 24 (39.3) | 18 (66.7)○ | 196 (46.4)○ | 0.0001* |
| **Duration** (days) | 5 [3–7] | 4 [2–7] | 5 [3–13] | 4 [2–7] | 8 [4–16] | 4.5 [3–11] | 5 [3–11] | 0.080 |
| **Hospital resources** | | | | | | | | |
| Hospital admission | 325 (77.0)●○ | 614 (87.8)● | 102 (84.3) | 132 (89.8)○ | 55 (90.2) | 22 (81.5) | 346 (81.9) | 0.0001* |
| Oxygen administration | 149 (35.3)●■ | 456 (65.2)●○■ | 79 (65.3)●○■ | 84 (57.1)●○■ | 40 (65.6)●○■ | 5 (18.5)■ | 184 (43.6)○ | 0.0001* |
| PICU admission | 71 (16.8)●○□ | 300 (42.9)● | 27 (22.3)●○ | 49 (33.3)□ | 28 (45.9)○ | 3 (11.1)●○ | 101 (23.9)●○ | 0.0001* |
| Mechanical ventilation | 19 (4.5)□ | 37 (5.3)○ | 14 (11.6)□ | 16 (10.8) | 10 (16.4) | 1 (3.7) | 60 (14.2)○ | 0.0001* |
| Total hospital stay (days) | 1.75 [1.1-2.7]● | 2.2 [1.4-3.6]●○□ | 3.0 [1.6-6.3]●○ | 2.5 [1.4-4.3]● | 3.2 [1.8-10.2]●□ | 2.8 [1.1-5.2] | 2.5 [1.3-5.5]● | 0.0001* |

Continuous variables are expressed as medians 25–75% interquartile ranges (IQR) and categorical data as frequency (%). Continuous variables were analyzed by analyzed by Kruskal-Wallis test followed by Holm post-hoc test. Categorical data were analyzed by $\chi^2$ test. CHD, congenital heart diseases; ICH, immunocompromised host; PICU, pediatric intensive care unit. Asterisks (*) indicate significant p values after applying the Bonferroni correction to adjust for multiple testing for families that included multiple parameters. Significant p values <0.001 are represented by filled symbols (●, ■, ▲, ◆); significant p values <0.05 are represented by blank symbols (○, □, △, ◇). Matching symbols indicate the groups for which significant differences were identified.

†Chronic respiratory conditions included: cystic fibrosis (CF) (n=18); chronic lung disease (CLD) (n=61); tracheoesophageal fistula, pulmonary hypertension, spontaneous pneumothorax, airway malacia, subglottic stenosis, cleft palate, and Pierre–Robin sequence.

§Others included: genetic syndromes (n=132); gastrointestinal diseases (n=68); neurological disorders (n=36); hematologic disease (n=17) and varied including endocrine, osteoarticular or genitourinary underlying diseases (n=169).

## Mortality and predictors *of* clinical outcomes

Thirty children died, all but two had major comorbidities including asthma/atopy (n=7), ICH (n=4), CHD (n=2), chronic respiratory conditions (CLD; n=1), prematurity (n=1) and other comorbidities (genetic syndromes, n=11; neuromuscular diseases, n=1; central diabetes insipidus, n=1). Most of these children (63.3%; 19/30) had high RV loads (≤25 $C_T$), intermediate (26–32 $C_T$) in 23.3% (7/30), and low (>32 $C_T$) in 4 (13.3%) children. A concomitant viral (n=6), bacterial (n=5) or fungal (n=1) co-infection was identified in 12 (40%) children. The per patient underlying condition, age, RV loads (ct values) and specific co-infection is included in S5 Table in S1 File.

Last, we performed multivariable analyses to define which comorbidities, adjusted by age, sex, RV loads and viral and bacterial co-infections, were associated with clinical outcomes defined as need for supplemental oxygen, PICU admission and longer length of stay, **Table 3**. Mortality was not included as an outcome due to the small sample size.

Of all comorbidities, asthma/atopy, prematurity and CHD were associated with increased odds of supplemental oxygen, PICU care and longer hospital stay. Chronic respiratory diseases were also independently associated with increase odds of oxygen supplementation and prolonged length of stay. Bacterial co-infections increased the risk of all three clinical outcomes, while older age and higher RV loads (<25 Ct values) were associated with increased odds of PICU care.

**Table 3. Adjusted risk factors associated with RV disease severity.**

| | Supplemental O2 | | PICU | | Length of stay | |
|---|---|---|---|---|---|---|
| | aOR (95% CI) | p-value | aOR (95% CI) | p-value | aOR (95% CI) | p-value |
| **Age** | | | | | | |
| Infants (<1yr) | **0.46 (0.31-0.68)** | **0.0001** | **0.43 (0.28-0.66)** | **0.0001** | **0.28 (0.18-0.44)** | **0.0001** |
| Toddlers (1–5yr) | 1.28 (0.85-1.93) | 0.233 | 1.23 (0.81-1.88) | 0.319 | **0.42 (0.27-0.65)** | **0.0001** |
| Children (>5–11yr) | 1.55 (0.98-2.47) | 0.060 | **1.66 (1.05-2.63)** | **0.029** | 0.65 (0.40-1.05) | 0.083 |
| **Sex (Female)** | 0.97 (0.79-1.18) | 0.775 | 0.99 (0.80-1.23) | 0.985 | 1.05 (0.86-1.28) | 0.626 |
| **Comorbidities** | | | | | | |
| Asthma/Atopy | **2.61 (2.00-3.41)** | **0.0001** | **2.82 (2.07-3.84)** | **0.0001** | **1.66 (1.26-2.19)** | **0.0001** |
| Chronic respiratory conditions † | **3.05 (1.96-4.75)** | **0.0001** | 0.97 (0.57-1.67) | 0.937 | **2.10 (1.34-3.28)** | **0.001** |
| Prematurity | **2.49 (1.68-3.71)** | **0.0001** | **2.52 (1.62-3.93)** | **0.0001** | **2.57 (1.72-3.86)** | **0.0001** |
| CHD | **3.00 (1.67-5.39)** | **0.0001** | **3.52 (1.95-6.35)** | **0.0001** | **2.62 (1.47-4.67)** | **0.001** |
| ICH | **0.22 (0.08-0.61)** | **0.004** | 0.31 (0.09-1.11) | 0.065 | 1.51 (0.64-3.53) | 0.341 |
| Others § | 1.24 (0.93-1.66) | 0.134 | 1.31 (0.92-1.86) | 0.133 | **1.96 (1.45-2.65)** | **0.0001** |
| **Viral co-detections** | 1.05 (0.84-1.33) | 0.760 | 1.16 (0.90-1.49) | 0.250 | 1.12 (0.89-1.42) | 0.309 |
| **Bacterial co-infections** | **2.06 (1.15-3.67)** | **0.001** | **5.43 (3.90-9.55)** | **0.0001** | **3.32 (1.77-6.22)** | **0.001** |
| **High RV loads (<25 Ct)** | 1.15 (0.94-1.41) | 0.162 | **1.40 (1.13-1.74)** | **0.002** | 1.12 (0.92-1.37) | 0.240 |

Data represented as adjusted odds ratios (aORs) and 95% confidence interval (CI). For age, children >11–21 yrs was used as a reference. For sex, male was used as reference. For comorbidities, healthy was used as a reference. For viral co-detections and bacterial co-infections, none was used as a reference. For high viral loads, intermediate and low (>25–40 Ct) were used as a reference. CHD, congenital heart diseases; ICH, immunocompromised host; O2, oxygen; PICU, pediatric intensive care unit. Length of stay was dichotomized by the median, and <2.1 days used as a refence.

†Chronic respiratory conditions included: cystic fibrosis (CF) (n = 18); chronic lung disease (CLD) (n = 61); tracheoesophageal fistula, pulmonary hypertension, spontaneous pneumothorax, airway malacia, subglottic stenosis, cleft palate, and Pierre–Robin sequence.

§Others included: genetic syndromes (n = 132); gastrointestinal diseases (n = 68); neurological disorders (n = 36); hematologic disease (n = 17) and varied including endocrine, osteoarticular or genitourinary underlying diseases (n = 169).

## Discussion

In this study including ~2,000 children and adolescents with RV ARI, we found that the majority had a chronic medical condition, and that the healthcare utilization, morbidity and even mortality was high and differed according to the type of comorbidity. Importantly, bacterial co-infections, older age and higher RV loads were also related to increased disease severity. To our knowledge this is the largest study conducted to date that analyzed the differential impact of symptomatic RV infections in children with existing comorbidities considering key factors that may have influenced clinical outcomes.

Studies have attributed lower rates of adverse outcomes among children with RV ARI which may be partially explained by the prolonged shedding of RV in the upper respiratory tract and the high rates of asymptomatic carriage [4,5]. In our study, we found a high prevalence of underlying comorbidities in children with RV ARI, with asthma/atopy and prematurity being the most common, which has been previously described [8,18–21]. We also found that the burden associated with RV ARI was high and that the need for hospitalization, supplemental oxygen, PICU admission or mechanical ventilation differed according to the type of underlying condition [8,19,22–27]. While ICH had the lowest rate of healthcare utilization, asthma/atopy, prematurity and CHD were consistently and independently associated with worse disease severity [8,11,18]. The lower rate of healthcare utilization in ICH was surprising and results should be viewed cautiously due to the limited number of ICH children included in the study. Nonetheless, differences in results between studies might be explained in part to the patients analyzed, as we only included children with symptomatic RV ARI [28–31]. Semiquantitative viral loads ($C_T$ values) were available for all children, and median RV loads were overall intermediate-high suggesting the potential contribution of RV to the clinical phenotype. Older age was also associated with worse clinical outcomes, which contrasts with other respiratory viruses such as RSV, which is typically linked to increased disease severity in young children [19,20,32,33].

Studies have reported rates of viral co-detection in children with RV ARI ranging from 15% to 50% depending on the study population and PCR platform used, a trend that has persisted post COVID-19 [8,20,21,34,35]. We found that the rates of RV/viral co-detections were common and ranged from 20% to 30%. These proportions were consistent across the different types of underlying conditions except for ICH in whom RV/viral co-detections were rare. While these results are intriguing, they need to be interpreted with caution as the number of ICH children included in the study was small, and the median age of these children high, which may partially explain the lower rates of RV/viral co-detections identified in these children. As shown by others we also found that the most common co-detected viruses were ADV, RSV and PIV [18,20,34]. Interestingly, the type of RV/viral co-detection differed based on the underlying condition, with RV/ADV being more common in premature and children with asthma/atopy, and RV/RSV in children with CHD and chronic respiratory diseases.

Bacterial co-infections in children with RV ARI are not well described, likely due to the low prevalence of bacteremia in children with LRTI or that lower respiratory or pleural fluid samples are often unavailable, yet antibiotics are frequently used. Studies have reported antibiotic use in <30% of children with RV ARI, which is slightly lower than the 40% found in our study. This likely related to the higher prevalence of children with comorbidities that required hospitalization including ICH, that may have biassed clinical decision making [27,36]. A case-control study of children with RV ARI found that bacterial co-infections were identified by sputum or throat cultures in ~70% of inpatients vs 1% of outpatients [21]. We identified bacterial co-infections in 2.9% of children, especially in ICH and those with chronic respiratory conditions. Our rates are similar to a study conducted in children requiring PICU care, that also defined bacterial co-infections according to lower respiratory tract or sterile site cultures [35]. In our study, 1.5% of children died, and all but two had chronic medical conditions, with 40% having a viral or bacterial co-infection. While bacterial co-infections were relatively infrequent in our cohort, our data suggest that coinfections may be associated with worse clinical outcomes and warrant the analysis of this parameter as possible contributor to RV disease severity in future studies.

This study has limitations. First, due to the retrospective study design, data collection was limited to information recorded in the EHR. However, to address missing or incomplete data, we analyzed objective outcomes, and two independent investigators manually reviewed and curated the data to ensure accuracy. The patient population represented a convenience sample, which may have introduced potential bias by over-representing the more severe forms of the disease. However, the study included children evaluated in the outpatient setting, and although viral testing was performed at the discretion of the attending physician, it aligned with hospital policy during the study period. The viral panel did not include viruses like endemic coronaviruses or bocaviruses, which may have played a role in disease presentation and outcomes. We, however, accounted for viral co-detections as well as RV loads in multivariable analyses and did not find a significant effect of these factors on clinical outcomes. RV typing was not performed, which limited our ability to assess the prevalence of RV-A, -B or -C infections in children with specific comorbidities and their contribution to clinical disease severity.

In summary, this study provides an in-depth characterization of the different comorbidities associated with severe RV infections in a large cohort of children and adolescents with RV ARI. It also highlights the potential role of RV loads, age or bacterial co-infections in RV-associated clinical outcomes. Defining the clinical phenotype of severe RV infections may help with resource allocation and patient management, with the ultimate goal of improving clinical outcomes.

## Supporting information

**S1 File. PLoS One_supplementary material_R2_F.**
(DOCX)

## Author contributions

**Conceptualization:** María Isabel Sánchez Códez, Asuncion Mejias.

**Data curation:** María Isabel Sánchez Códez, Katherine Moyer, Asuncion Mejias.

**Formal analysis:** María Isabel Sánchez Códez, Isabel Benavente Fernandez, Asuncion Mejias.

**Funding acquisition:** Asuncion Mejias.

**Investigation:** María Isabel Sánchez Códez, Isabel Benavente Fernandez, Katherine Moyer, Amy L. Leber, Octavio Ramilo, Asuncion Mejias.

**Methodology:** María Isabel Sánchez Códez, Isabel Benavente Fernandez, Katherine Moyer, Amy L. Leber, Octavio Ramilo, Asuncion Mejias.

**Project administration:** María Isabel Sánchez Códez, Asuncion Mejias.

**Resources:** Amy L. Leber, Asuncion Mejias.

**Software:** Asuncion Mejias.

**Visualization:** Asuncion Mejias.

**Writing – original draft:** María Isabel Sánchez Códez.

**Writing – review & editing:** María Isabel Sánchez Códez, Isabel Benavente Fernandez, Katherine Moyer, Amy L. Leber, Octavio Ramilo, Asuncion Mejias.

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
