## [Decision Letter · Decision Letter 0]

5 Dec 2024

PONE-D-24-47497The differential burden of acute rhinovirus infections in children with underlying conditionsPLOS ONE

Dear Dr. Mejias,

Thank you for submitting your manuscript to PLOS ONE. After careful consideration, we feel that it has merit but does not fully meet PLOS ONE’s publication criteria as it currently stands. Therefore, we invite you to submit a revised version of the manuscript that addresses the points raised during the review process.

Thank you for submitting your manuscript titled *"The Differential Burden of Acute Rhinovirus Infections in Children with Underlying Conditions."* This study provides valuable insights into how rhinovirus (RV) infections affect children with comorbidities, particularly highlighting key predictors of disease severity. The research is well-conducted and addresses an important topic; however, there are several areas where revisions could improve the clarity, depth, and presentation of your work. See below for comments:

**Major comments:**

1) The inclusion of young adults aged 18–21 years needs further explanation. It’s not immediately clear how this age group fits within a study focused on pediatric populations, and justification would help align the cohort with the study objectives. Similarly, the approach to excluding recurrent visits only within the same calendar year raises questions. A stricter time window, such as excluding visits occurring within 14–28 days, might provide a more accurate picture of unique episodes while avoiding potential confounding from closely spaced interactions.

2) The way comorbidities are classified also require further clarification. When participants had multiple conditions, the rationale for prioritizing one condition over others should be clearly conveyed as this will ensure transparency and reproducibility. Additionally, more details are required for the statistical analyses section. For example, information on post-hoc tests, corrections for multiple comparisons, and how continuous variables like length of stay were dichotomized will provide stronger justification.

3) There needs to be consistency in terminology used throughout the manuscript. “Asthma/Atopy” is inconsistently labeled in tables and text. Also, terms like “CF” need to be clearly defined at first mention and subsequently abbreviated where appropriate.

4) The results related to bacterial and fungal pathogens need to be better integrated. Further clarification on the methodology for pathogen detection, including subgroup analysis is required too.

5) The unexpectedly low healthcare utilization and co-infection rates in immunocompromised children require more discussion which is inadequate at present.

6) The figures and tables are well-prepared but can benefit from added indicators of statistical significance, such as asterisks or p-values. In cases where post-hoc tests yielded significant results, this should be noted in the tables; if none were significant, this should also be explicitly stated to avoid ambiguity.

7) Your conclusions are compelling but occasionally overstated, ie: the relationship between high RV loads and severe outcomes should be presented as an association rather than causation. Similarly, your claims about bacterial co-infections influencing triage models should be revisited, given their relatively low prevalence in your assessed cohort.

**Minor comments:**

1) The manuscript contains occasional grammatical errors and typographical inconsistencies: On Page 2 Line 2, it should read “*are a well-known trigger* ,” and Page 15 Line 26 contains a redundant “*we* ” that should be removed. Reference formatting also needs revision to ensure consistency of article titles.

2) Reviewing and revising figure captions for clarity and precision would further improve presentation.

3) A subsection in the Discussion to explicitly address limitations such as the retrospective design, potential sampling bias, and absence of RV molecular typing will significantly strengthen the manuscript and allow for a more balanced interpretation of the findings.

This study offers a significant contribution to understanding RV infections in children with comorbidities, and we appreciate the effort involved in its preparation. Addressing these revisions will enhance the manuscript's clarity, consistency, and impact. We look forward to your revised submission.

We look forward to receiving your revised manuscript.

Kind regards,

Kevin Looi, Ph.D

Academic Editor

PLOS ONE

Journal requirements: When submitting your revision, we need you to address these additional requirements. 1. Please ensure that your manuscript meets PLOS ONE's style requirements, including those for file naming. The PLOS ONE style templates can be found at https://journals.plos.org/plosone/s/file?id=wjVg/PLOSOne_formatting_sample_main_body.pdf and https://journals.plos.org/plosone/s/file?id=ba62/PLOSOne_formatting_sample_title_authors_affiliations.pdf 2. We note that you have indicated that there are restrictions to data sharing for this study. For studies involving human research participant data or other sensitive data, we encourage authors to share de-identified or anonymized data. However, when data cannot be publicly shared for ethical reasons, we allow authors to make their data sets available upon request. For information on unacceptable data access restrictions, please see http://journals.plos.org/plosone/s/data-availability#loc-unacceptable-data-access-restrictions.  Before we proceed with your manuscript, please address the following prompts: a) If there are ethical or legal restrictions on sharing a de-identified data set, please explain them in detail (e.g., data contain potentially identifying or sensitive patient information, data are owned by a third-party organization, etc.) and who has imposed them (e.g., a Research Ethics Committee or Institutional Review Board, etc.). Please also provide contact information for a data access committee, ethics committee, or other institutional body to which data requests may be sent. b) If there are no restrictions, please upload the minimal anonymized data set necessary to replicate your study findings to a stable, public repository and provide us with the relevant URLs, DOIs, or accession numbers. Please see http://www.bmj.com/content/340/bmj.c181.long for guidelines on how to de-identify and prepare clinical data for publication. For a list of recommended repositories, please see https://journals.plos.org/plosone/s/recommended-repositories. You also have the option of uploading the data as Supporting Information files, but we would recommend depositing data directly to a data repository if possible. Please update your Data Availability statement in the submission form accordingly. 3. Your ethics statement should only appear in the Methods section of your manuscript. If your ethics statement is written in any section besides the Methods, please move it to the Methods section and delete it from any other section. Please ensure that your ethics statement is included in your manuscript, as the ethics statement entered into the online submission form will not be published alongside your manuscript.  4. Please include captions for your Supporting Information files at the end of your manuscript, and update any in-text citations to match accordingly. Please see our Supporting Information guidelines for more information: http://journals.plos.org/plosone/s/supporting-information. 

Reviewers' comments:

Reviewer's Responses to Questions

**Comments to the Author**

1. Is the manuscript technically sound, and do the data support the conclusions?

Reviewer #1: Yes

2. Has the statistical analysis been performed appropriately and rigorously? 

Reviewer #1: Yes

3. Have the authors made all data underlying the findings in their manuscript fully available?

Reviewer #1: Yes

4. Is the manuscript presented in an intelligible fashion and written in standard English?

Reviewer #1: Yes

5. Review Comments to the Author

Reviewer #1: This paper describes a study of children presenting to either hospital or outpatient services with symptomatic RV infection. The authors aimed to identify clinical factors and co-morbid conditions associated with increased care needs. They found that comorbidities were common in this cohort, with asthma, prematurity and chronic heart disease all associated with supplemental oxygen need, PICU admission and increased length of stay. The paper is overall well-written and the results provide important insight into the importance of considering comorbid conditions when treating ARI; however, it lacks clarity in some areas that, when addressed, will aid the reader in their ability to interpret and assess the quality of the data presented.

Major comments

Page 3 Methods: What was the rationale for including young adults aged 18-21 years in this study? Also, why were subsequent hospital presentations/outpatient interactions only excluded from the study if they took place in the same calendar year? Depending on the precise aim, including either only one interaction per participant, or all interactions excluding those occurring very close together (14 or 28 days are common cut-offs), would be preferable.

Page 4 Lines 7-8: The reference to cystic fibrosis renders this entire sentence confusing with the current wording. Please clarify which samples were used for bacterial testing and if necessary, the sub-groups of children the samples were available in.

Page 4 Lines 12-14: Please provide the hierarchical order of comorbidities in full.

Page 4 Statistical Analysis: Some details of the analyses are reported in the text accompanying the tables but not in the Methods text, such as the post-hoc tests used (also, multiple test correction for post-hoc tests is not mentioned in either section) and the metric used to convert length of stay into a binary suitable for logistic regression. Please include this information here.

Page 5 Results: Similarly to the previous comment, some results are only reported in the footnotes to the tables but not in the body text, e.g. the other respiratory diseases besides asthma and “other” category diagnoses.

Page 7 Table 1: The “asthma” category is labelled in this table as “Asthma/Atopy”. This needs to made consistent between the body text and table. Also, why are lymphocytes reported as a percentage but total white cells and neutrophils are reported as absolute counts (albeit not on the same scale)? Also, were any of the post-hoc tests significant? If so, can this please be indicated in the table, and if not, can it please be stated in the text?

Page 9: As per the comment regarding Page 4, what does “CF” refer to in this context?

Page 10: Fungal pathogens are not mentioned in the Methods, making their appearance here sudden and somewhat confusing.

Page 11 Table 2: The “asthma” category is labelled in this table as “Asthma/Atopy”. This needs to made consistent between the body text and table. Also, were any of the post-hoc tests significant? If so, can this please be indicated in the table, and if not, can it please be stated in the text?

Page 12 Results: When describing which comorbidities appear to drive differences, please indicate if post-hoc testing was performed or if this is purely a numeric assessment.

Page 14 Discussion: Can the authors comment on the (perhaps unusually) low healthcare needs and co-infection rates observed in immunocompromised children?

Figures 1 and 2: Can indicators of significance please be added to these figures?

Minor comments

Page 2 Line 2: Correct to “…are a well-known trigger…”

Page 2 Line 3: Correct to “known.”

Page 2 Line 7: Correct to “…outcomes were compared…”

Page 2 Line 14: Replace “while” with “whereas”.

Page 2 Line 15: Correct to “…bacterial co-infections as consistently associated…”

Page 3 Line 3: Correct to “…role of RVs in respiratory disease…”

Page 3 Line 16: Correct to “This was a retrospective study of…”

Page 3 Line 18: Correct to “…clinical data was extracted…”

Page 4 Line 5: Correct to “…detection of one of the other viruses…”

Page 4 Line 9: Correct to “…analysis purposes…”

Page 4 Line 12: Please omit the “(7)”.

Page 4 Line 23: Correct to “…data were not normally distributed.”

Page 8 Line 4: Correct to “depicts”.

Page 8 Line 6: Correct to “The percentages at the top of the bars…”

Page 8 Line 7: Correct to “…those at the…”

Page 8 Line 19: Correct to “diagnoses”.

Page 9 Line 11: Correct to “RV/ADV co-detection was more…”

Page 9 Line 12: Replace “while” with “whereas”.

Page 9 Line 15: Correct to “…in ICH was rare…”

Page 9 Line 16: Change to “More than one additional respiratory virus…”

Page 12 Line 24: Correct to “comorbidities”.

Page 12 Line 25: Correct to “co-infection, were associated…”

Page 13 Line 5: Correct to “associated with increased odds of…”

Page 14 Line 7: Correct to “…with RV ARI as likely…”

Page 15 Line 15: Please reword “Nevertheless, those and our study…” as it is currently unclear what “those” refers to.

Page 15 Line 17: Correct to “contributor to RV disease severity…”

Page 15 Line 18: Insert comma for “…study design, data collection was…”

Page 15 Line 26: Omit second “we”.

Page 17 References: The capitalisation of article titles is inconsistent between references. Can this please be made consistent in accordance with the citation style guidelines?

6. PLOS authors have the option to publish the peer review history of their article (what does this mean? ). If published, this will include your full peer review and any attached files.

**Do you want your identity to be public for this peer review?** For information about this choice, including consent withdrawal, please see our Privacy Policy .

Reviewer #1: No

---

## [Author Response · Author response to Decision Letter 1]

27 Jan 2025

EDITOR’S COMMENTS TO AUTHOR:

Thank you for submitting your manuscript titled "The Differential Burden of Acute Rhinovirus Infections in Children with Underlying Conditions." This study provides valuable insights into how rhinovirus (RV) infections affect children with comorbidities, particularly highlighting key predictors of disease severity. The research is well-conducted and addresses an important topic; however, there are several areas where revisions could improve the clarity, depth, and presentation of your work. See below for comments:

Major comments:

Comment #1. The inclusion of young adults aged 18–21 years needs further explanation. It’s not immediately clear how this age group fits within a study focused on pediatric populations, and justification would help align the cohort with the study objectives. Similarly, the approach to excluding recurrent visits only within the same calendar year raises questions. A stricter time window, such as excluding visits occurring within 14–28 days, might provide a more accurate picture of unique episodes while avoiding potential confounding from closely spaced interactions.

Response #1: Thanks for the questions, we appreciate the opportunity to clarify these aspects. We included patients up to 21 years of age since this is the catchment population that is cared for at Nationwide Children’s Hospital (NCH). It is not uncommon for Children’s Hospitals in the United States to provide care through age 21 as recommended by the American Academy of Pediatrics that published years ago and has reaffirmed that the upper age limits of pediatrics is 21 years [1]. We now clarified this aspect in the material and methods section.

Regarding the exclusion of recurrent visits, the editor is correct. We indeed excluded patients who had another visit not only 14-28 days after the initial encounter, but during that calendar year. We have clarified these aspects in the material and methods section as well. We recognize that we may have missed different episodes of rhinovirus infection in the same patient over a year, but we opted to be conservative to avoid the overrepresentation of certain patient populations.

Comment #2. The way comorbidities are classified also require further clarification. When participants had multiple conditions, the rationale for prioritizing one condition over others should be clearly conveyed as this will ensure transparency and reproducibility. Additionally, more details are required for the statistical analyses section. For example, information on post-hoc tests, corrections for multiple comparisons, and how continuous variables like length of stay were dichotomized will provide stronger justification.

Response #2: Thanks much for your comments and suggestions. For patients with more than one underlying condition we prioritize the comorbidity associated with an increased risk for severe RV infection based on previous studies [2-6]; i.e if a child had asthma and renal disease that child was categorized under asthma. We have now clarified this priority classification under the material and methods section.

Following reviewers’ suggestions, we have also clarified in detail our statistical approach. Briefly, for the analyses of continuous variables we used Kruskal-Wallis followed by Holm’s post-hoc test to adjust for multiple comparisons. In addition, the Bonferroni correction was applied to correct for multiple testing when analyzing families that included multiple parameters such as diagnoses, medications, or underlying conditions. The asterisks (*) included in Tables 1-2 and Tables S2-S4 indicate the significant adjusted p-values.

To facilitate analyses interpretation, we dichotomized length of stay by the median which was 2.1 days. We have now added this information to the material and methods section and Table 2 footnote.

Comment #3. There needs to be consistency in terminology used throughout the manuscript. “Asthma/Atopy” is inconsistently labeled in tables and text. Also, terms like “CF” need to be clearly defined at first mention and subsequently abbreviated where appropriate.

Response #3. Apologies for the inconsistencies. We have thoroughly reviewed the text, and we use now the terminology consistently. In addition, abbreviations are spelled out the first time they appear in the text.

Comment #4. The results related to bacterial and fungal pathogens need to be better integrated. Further clarification on the methodology for pathogen detection, including subgroup analysis is required too.

Response #4. Thanks for the suggestions. We have now included the methodology used for pathogen detection under material and methods, which were all based on culture. In addition, we have reworded and integrated the findings related to bacterial and fungal pathogens as suggested.

Comment #5. The unexpectedly low healthcare utilization and co-infection rates in immunocompromised children require more discussion which is inadequate at present.

Response #5. We totally agree with the reviewer as we were also surprised. However, the number of ICH children was very small, and the median age of these children high, which may partially explain these results, and limits the generalizability of the results. We now mention these aspects in the discussion.

Comment #6. The figures and tables are well-prepared but can benefit from added indicators of statistical significance, such as asterisks or p-values. In cases where post-hoc tests yielded significant results, this should be noted in the tables; if none were significant, this should also be explicitly stated to avoid ambiguity.

Response #6. Thanks for the comment. Following the editor’s suggestion, to facilitate the identification of the groups that are different after post-hoc analyses we have included different symbols, both in the tables (main and supplementary) and figures.

Comment #7: Your conclusions are compelling but occasionally overstated, ie: the relationship between high RV loads and severe outcomes should be presented as an association rather than causation. Similarly, your claims about bacterial co-infections influencing triage models should be revisited, given their relatively low prevalence in your assessed cohort.

Response #7. Thanks for the comment. We have toned down our conclusions to reflect the descriptive nature of the study and that results needs to be interpreted with caution due to the low prevalence of some of the covariates.

Minor comments:

Comment #8. The manuscript contains occasional grammatical errors and typographical inconsistencies: On Page 2 Line 2, it should read “are a well-known trigger,” and Page 15 Line 26 contains a redundant “we” that should be removed. Reference formatting also needs revision to ensure consistency of article titles.

Response #8. Apologies for the oversight. We have thoroughly reviewed the manuscript for inconsistencies and reformatted the references.

Comment #9. Reviewing and revising figure captions for clarity and precision would further improve presentation

Response #9. Figures legends have been reviewed and updated.

Comment #10. A subsection in the Discussion to explicitly address limitations such as the retrospective design, potential sampling bias, and absence of RV molecular typing will significantly strengthen the manuscript and allow for a more balanced interpretation of the findings.

Response #10: reviewer is correct. We have now included in the discussion a section highlighting the study limitations including the retrospective study design or lack of RV molecular typing among others.

This study offers a significant contribution to understanding RV infections in children with comorbidities, and we appreciate the effort involved in its preparation. Addressing these revisions will enhance the manuscript's clarity, consistency, and impact. We look forward to your revised submission.

REVIEWER #1 COMMENTS TO THE AUTHOR

This paper describes a study of children presenting to either hospital or outpatient services with symptomatic RV infection. The authors aimed to identify clinical factors and co-morbid conditions associated with increased care needs. They found that comorbidities were common in this cohort, with asthma, prematurity and chronic heart disease all associated with supplemental oxygen need, PICU admission and increased length of stay. The paper is overall well-written, and the results provide important insight into the importance of considering comorbid conditions when treating ARI; however, it lacks clarity in some areas that, when addressed, will aid the reader in their ability to interpret and assess the quality of the data presented.

Major comments

Comment #1: Page 3 Methods: What was the rationale for including young adults aged 18-21 years in this study? Also, why were subsequent hospital presentations/outpatient interactions only excluded from the study if they took place in the same calendar year? Depending on the precise aim, including either only one interaction per participant, or all interactions excluding those occurring very close together (14 or 28 days are common cut-offs), would be preferable.

Response #1: Please see response #1 to the editor. Briefly Is not uncommon for Children’s Hospitals in the United States to provide care through 21 years of ager and even older in certain situations. Thus, we included patients up to 21 years of age since this is the catchment population that is cared for at Nationwide Children’s Hospital (NCH). We have now clarified this aspect in the material and methods section. In fact, the American Academy of Pediatrics published years ago and has reaffirmed that the upper age limits of pediatrics is 21 years [1].

Regarding the exclusion of recurrent visits, reviewer #1 is correct. We indeed excluded patients who had another visit not only 14-28 days after the initial encounter, but during that year. We have clarified these aspects in the material and methods section as well. We recognize that we may have missed episodes of rhinovirus infection in the same patient over that year, but we opted to be conservative to avoid the overrepresentation of certain patient populations, as rhinovirus can be shed for extended periods of time.

Comment #2. Page 4 Lines 7-8: The reference to cystic fibrosis renders this entire sentence confusing with the current wording. Please clarify which samples were used for bacterial testing and if necessary, the sub-groups of children the samples were available in.

Response #2: We have reworded the sentence where we mention cystic fibrosis (CF) for clarity. Regarding bacterial testing and coinfections, bacterial and fungal testing was performed by culture per standard of care. Pathogens were included in the study when isolated from blood, pleural fluid, or lower respiratory tract samples. In addition, in symptomatic children with CF, throat cultures were considered a valid sample to assess bacterial coinfections. This has been now clarified in the Methods section.

Comment #3: Page 4 Lines 12-14: Please provide the hierarchical order of comorbidities in full.

Response #3: Please see response #2 to the editor. Briefly, for patients with more than one underlying condition we prioritize the comorbidity that associated with an increased risk for severe RV infection based on previous studies [2-6]; i.e if a child had asthma and renal disease that child was categorized under asthma. Please find below the hierarchical order that we used below, which is also included now under material and methods: (1) Immunocompromised host (ICH); (2) Asthma/atopy ; (3) Other chronic respiratory conditions including CF or chronic lung disease (CLD); (4) prematurity without CLD; (5) congenital heart disease (CHD) and (6) Other conditions (i.e endocrine, renal, etc.).

Comment #4. Page 4 Statistical Analysis: Some details of the analyses are reported in the text accompanying the tables but not in the Methods text, such as the post-hoc tests used (also, multiple test correction for post-hoc tests is not mentioned in either section) and the metric used to convert length of stay into a binary suitable for logistic regression. Please include this information here.

Response #4: Please see response #2 to the editor. Briefly for the analyses of continuous variables we used Kruskal-Wallis followed by Holm’s post-hoc test to adjust for multiple comparisons. For categorical data, we used the Chi-squared test. In addition, the Bonferroni correction was applied to correct for multiple testing when analyzing families that included multiple parameters such as diagnoses, medications, or underlying conditions. The asterisks (*) included in Tables 1-2 and Tables S2-S4 indicate the significant adjusted p-values.

As done in previous studies we dichotomized length of stay by the median which was 2.1 days. We have now added this information to the material and Methods section.

Comment #5. Page 5 Results: Similarly to the previous comment, some results are only reported in the footnotes to the tables but not in the body text, e.g. the other respiratory diseases besides asthma and “other” category diagnoses.

Response #5. We apologize for the oversight. We have now ensured that all relevant results are described in the main text, rather than just in the footnotes of the tables. See updated sections (page 5, lines 19-25).

Comment #6: Page 7 Table 1: The “asthma” category is labelled in this table as “Asthma/Atopy”. This needs to be made consistent between the body text and table. Also, why are lymphocytes reported as a percentage, but total white cells and neutrophils are reported as absolute counts (albeit not on the same scale)? Also, were any of the post-hoc tests significant? If so, can this please be indicated in the table, and if not, can it please be stated in the text?

Response #6: Apologies for the mislabeling. We have carefully reviewed the manuscript to ensure that we use consistently the terminology. The terms "asthma" and "atopy" are now used uniformly throughout the text, tables and figures. We have also updated Table 1 and lymphocytes are now reported as absolute lymphocyte counts (ALC), consistent with the reporting of total white cells and neutrophils (page 7, Table 1). Significant post-hoc tests are now included in the tables (please see the different symbols included).

Comment #7: Page 9: As per the comment regarding Page 4, what does “CF” refer to in this context?

Response #7: Thanks for allowing us to clarify this. “CF” refers to throat cultures from patients with cystic fibrosis (CF). We have changed the text accordingly (page 10, line 14).

Comment #8: Page 10: Fungal pathogens are not mentioned in the Methods, making their appearance here sudden and somewhat confusing.

Response #8: Reviewer is correct. We have now added a brief sentence about the different types of cultures, including those for fungal pathogens, in the methods section.

Comment #9: Page 11 Table 2: The “asthma” category is labelled in this table as “Asthma/Atopy”. This needs to be made consistent between the body text and table. Also, were any of the post-hoc tests significant? If so, can this please be indicated in the table, and if not, can it please be stated in the text?

Response #9: Please see response #3 to the editor and response to comment #6 above. We have reviewed the manuscript and tables and use now asthma/atopy consistently. In addition, the significant post-hoc tests are included in Table 2.

Comment #10: Page 12 Results: When describing which comorbidities appear to drive differences, please indicate if post-hoc testing was performed or if this is purely a numeric assessment.

Response #10: Thanks for allowing is to clarify this aspect. Results indicated statistically significant differences between comorbidities adjusted for multiple testing. These has been clarified in the text and Table 2 footnote.

Comment #11: Page 14 Discussion: Can the authors comment on the (perhaps unusually) low healthcare needs and co-infection rates observed in immunocompromised children?

Response #11: We agree with the reviewer that this is an interesting observation. In our study, however only 27 children who were immunocompromised hosts (ICH)

---

## [Decision Letter · Decision Letter 1]

23 Feb 2025

PONE-D-24-47497R1The differential burden of acute rhinovirus infections in children with underlying conditionsPLOS ONE

Dear Dr. Mejias,

Thank you for submitting your manuscript to PLOS ONE. After careful consideration, we feel that it has merit but does not fully meet PLOS ONE’s publication criteria as it currently stands. Therefore, we invite you to submit a revised version of the manuscript that addresses the points raised during the review process.

We look forward to receiving your revised manuscript.

Kind regards,

Tai-Heng Chen, M.D., Ph.D.

Academic Editor

PLOS ONE

Journal Requirements:

Reviewers' comments:

Reviewer's Responses to Questions

**Comments to the Author**

1. If the authors have adequately addressed your comments raised in a previous round of review and you feel that this manuscript is now acceptable for publication, you may indicate that here to bypass the “Comments to the Author” section, enter your conflict of interest statement in the “Confidential to Editor” section, and submit your "Accept" recommendation.

Reviewer #1: All comments have been addressed

2. Is the manuscript technically sound, and do the data support the conclusions?

Reviewer #1: Yes

3. Has the statistical analysis been performed appropriately and rigorously? 

Reviewer #1: Yes

4. Have the authors made all data underlying the findings in their manuscript fully available?

Reviewer #1: Yes

5. Is the manuscript presented in an intelligible fashion and written in standard English?

Reviewer #1: Yes

6. Review Comments to the Author

Reviewer #1: (No Response)

7. PLOS authors have the option to publish the peer review history of their article (what does this mean? ). If published, this will include your full peer review and any attached files.

**Do you want your identity to be public for this peer review?** For information about this choice, including consent withdrawal, please see our Privacy Policy .

Reviewer #1: No

---

## [Author Response · Author response to Decision Letter 2]

2 Mar 2025

Reviewers did not have specific questions or comments for us to address in this version of the manuscript (R2). We thank them for their previous comments and suggestions (R1) that had greatly improved our manuscript.

Journal Requirements:

Please review your reference list to ensure that it is complete and correct. If you have cited papers that have been retracted, please include the rationale for doing so in the manuscript text or remove these references and replace them with relevant current references. Any changes to the reference list should be mentioned in the rebuttal letter that accompanies your revised manuscript. If you need to cite a retracted article, indicate the article’s retracted status in the References list and also include a citation and full reference for the retraction notice.

Response:

We have thoroughly reviewed our reference list and have made minor changes that are highlighted in the revised version of the manuscript. Specifically:

Introduction:

• Reference #6 which was previously located in page 3, line 6 has been moved to the previous line (line 5, page 3), as it was originally misplaced.

Discussion

• Page 13, line 1: We have updated the three previous references (10, 28 and 29) and are now #8, 11 and 18. The three updated references reflect better the content of the discussion.

• Page 13, lines 4 and 5: we have added three new references, 28, 29 and 31 and mention a previously cited reference (#30) that was relevant to the topic discussed in that section.

• Page 13, line 8, reference 33 has been moved to this section that fits better with the content of the sentence.

PONE-D-24-47497R1

The differential burden of acute rhinovirus infections in children with underlying conditions

PLOS ONE

Dear Editor, as requested we have included the following items in the revised version of the manuscript:

It appears that the last resubmission (R1) satisfied all previous comments and concerns raised by the reviewers and editors. There were no specific comments for us to address in the manuscript version that we received last week. Please see below Reviewer #1 comments.

There were only changes requested regarding the references, which we have updated. These references are now highlighted in yellow.

This has been included

No changes have occurred in our COI

Comments to the Author

1. If the authors have adequately addressed your comments raised in a previous round of review and you feel that this manuscript is now acceptable for publication, you may indicate that here to bypass the “Comments to the Author” section, enter your conflict of interest statement in the “Confidential to Editor” section, and submit your "Accept" recommendation.

Reviewer #1: All comments have been addressed

2. Is the manuscript technically sound, and do the data support the conclusions?

Reviewer #1: Yes

3. Has the statistical analysis been performed appropriately and rigorously?

Reviewer #1: Yes

4. Have the authors made all data underlying the findings in their manuscript fully available?

Reviewer #1: Yes

5. Is the manuscript presented in an intelligible fashion and written in standard English?

Reviewer #1: Yes

6. Review Comments to the Author

Reviewer #1: (No Response)

7. PLOS authors have the option to publish the peer review history of their article (what does this mean?). If published, this will include your full peer review and any attached files.

Do you want your identity to be public for this peer review? For information about this choice, including consent withdrawal, please see our Privacy Policy.

Reviewer #1: No

---

## [Decision Letter · Decision Letter 2]

4 Apr 2025

The differential burden of acute rhinovirus infections in children with underlying conditions

PONE-D-24-47497R2

Dear Dr. Mejias,

We’re pleased to inform you that your manuscript has been judged scientifically suitable for publication and will be formally accepted for publication once it meets all outstanding technical requirements.

Kind regards,

Tai-Heng Chen, M.D., Ph.D.

Academic Editor

PLOS ONE

Reviewers' comments:

Reviewer's Responses to Questions

**Comments to the Author**

1. If the authors have adequately addressed your comments raised in a previous round of review and you feel that this manuscript is now acceptable for publication, you may indicate that here to bypass the “Comments to the Author” section, enter your conflict of interest statement in the “Confidential to Editor” section, and submit your "Accept" recommendation.

Reviewer #1: All comments have been addressed

2. Is the manuscript technically sound, and do the data support the conclusions?

Reviewer #1: (No Response)

3. Has the statistical analysis been performed appropriately and rigorously? 

Reviewer #1: (No Response)

4. Have the authors made all data underlying the findings in their manuscript fully available?

Reviewer #1: (No Response)

5. Is the manuscript presented in an intelligible fashion and written in standard English?

Reviewer #1: (No Response)

6. Review Comments to the Author

Reviewer #1: (No Response)

7. PLOS authors have the option to publish the peer review history of their article (what does this mean? ). If published, this will include your full peer review and any attached files.

**Do you want your identity to be public for this peer review?** For information about this choice, including consent withdrawal, please see our Privacy Policy .

Reviewer #1: No

---

## [Editor Report · Acceptance letter]

PONE-D-24-47497R2

PLOS ONE

Dear Dr. Mejias,

I'm pleased to inform you that your manuscript has been deemed suitable for publication in PLOS ONE. Congratulations! Your manuscript is now being handed over to our production team.

Kind regards,

on behalf of

Dr. Tai-Heng Chen

Academic Editor

PLOS ONE